# Epidemiology of extended-spectrum beta-lactamase-producing *Escherichia coli* at the human-animal-environment interface in a farming community of central Uganda

**James Muleme**[1,2]*, **David Musoke**[1], **Bonny E. Balugaba**[1], **Stevens Kisaka**[2,3], **Frederick E. Makumbi**[3], **Esther Buregyeya**[1], **John Bosco Isunju**[1], **Rogers Wambi**[2,4], **Richard K. Mugambe**[1], **Clovice Kankya**[2], **Musso Munyeme**[5], **John C. Ssempebwa**[1]

**1** Department of Disease Control and Environmental Health, Makerere University School of Public Health, Kampala, Uganda, **2** Department of Biosecurity Ecosystems and Veterinary Public Health, Makerere University College of Veterinary Medicine Animal Resources and Biosecurity, Kampala, Uganda, **3** Department of Epidemiology and Biostatistics, Makerere University School of Public Health, Kampala, Uganda, **4** Clinical Laboratories, Mulago National Referral Hospital, Kampala, Uganda, **5** Department of Disease Control, University of Zambia, Lusaka, Zambia

* mulemej@gmail.com

## Abstract

### Background

Extended-spectrum beta-lactamase-producing *Escherichia coli* (ESBL-Ec) represents a significant global public health concern. The epidemiology of ESBL-Ec in Uganda is not well understood although it is harbored by humans, animals, and the environment. This study explains the epidemiology of ESBL-Ec using a one health approach in selected farming households in Wakiso district, Uganda.

### Methodology

Environmental, human, and animal samples were collected from 104 households. Additional data were obtained using observation checklists and through interviews with household members using a semi-structured questionnaire. Surface swabs, soil, water, human and animal fecal samples were introduced onto ESBL chromogenic agar. The isolates were identified using biochemical tests and double-disk synergy tests. To assess associations, prevalence ratios (PRs) were computed using a generalized linear model (GLM) analysis with modified Poisson and a log link with robust standard errors in R software.

### Results

Approximately 83% (86/104) households had at least one positive ESBL-Ec isolate. The overall prevalence of ESBL-Ec at the human-animal-environment interface was approximately 25.0% (95% CI: 22.7–28.3). Specifically, humans, animals and the environment had an ESBL-Ec prevalence of 35.4%, 55.4%, and 9.2% respectively. Having visitors (adj PR = 1.19, 95% CI: 1.04–1.36), utilizing veterinary services (adj PR = 1.39, 95% CI: 1.20–1.61)

**Data Availability Statement:** All relevant data are within the paper and its Supporting Information files.

**Funding:** JM was supported by the Consortium for Advanced Research Training in Africa (CARTA). CARTA is jointly led by the African Population and Health Research Center and the University of the Witwatersrand and funded by the Carnegie Corporation of New York (Grant No. G-19-57145), Swedish International Development Cooperation Agency (Sida) (Grant No:54100113), Uppsala Monitoring Center, Norwegian Agency for Development Cooperation (Norad), and by the Wellcome Trust [reference no. 107768/Z/15/Z] and the United Kingdom Foreign, Commonwealth & Development Office, with support from the Developing Excellence in Leadership, Training and Science in Africa (DELTAS Africa) programme. In addition, the Climate Change and Infectious Diseases: One Health Approach (CIDIMOH) project under the Norwegian Programme for Capacity Development in Higher Education and Research for Development (NORHED II) supported JM to carry out part of the laboratory component during sample analysis. The statements made and views expressed are solely the responsibility of the Fellow. The funders had no role in study design, data collection and analysis, decision to publish, or preparation of the manuscript.

**Competing interests:** The authors have declared that no competing interests exist.

and using animal waste for gardening (adj PR = 1.29, 95% CI: 1.05–1.60) were positively associated with household ESBL-Ec contamination. Covering the drinking water container with a lid (adj PR = 0.84 95% CI: 0.73–0.96) was associated with absence of ESBL-Ec in a household.

## Conclusion

There is wider dissemination of ESBL-Ec in the environment, humans, and animals, indicating poor infection prevention and control (IPC) measures in the area. Improved collaborative one health mitigation strategies such as safe water chain, farm biosecurity, household and facility-based IPC measures are recommended to reduce the burden of antimicrobial resistance at community level.

## Introduction

If no appropriate measures are taken, it is projected that approximately 10,000,000 deaths and about US$100 trillion in economic losses will occur per year by 2050 due to antimicrobial resistance [1]. Infections caused by Extended Beta Lactamase producing Escherichia coli (ESBL-Ec) have been implicated in severe disease outbreaks globally [2–4]. Studies have reported that the resistance to third generation cephalosporin in *E. coli* ranged from 0–87% whereas resistance to fluoroquinolones ranged from 0–98%. In Tanzania, deaths due to sepsis among neonates have been attributed to ESBL infections [5]. A review by Martischang et al, in 2020, reported that the prevalence of ESBL-Ec co-carriage among household members ranged from 8% to 37% [6–8]. This implies that such infections are emerging "One Health" threats compromising the safety and health of humans, animals, as well as the purity of the environment globally. Even though irrational drug use has been implicated as an internationally recognized cause of intrinsic antimicrobial resistance (AMR) [7], the interactions and associated dynamics among humans, animals and environment are increasingly being pinned as major phenomena for the registered global AMR burden [6, 9]. To understand the dynamics of the dispersal of ESBL-Ec into natural environments beyond human and domestic animal population, it is important to keep in mind the general *E. coli* population as well. *E. coli* is ubiquitous, and asymptomatically colonizes the gut of birds and mammals.

The cause of human ESBL-Ec colonization is still contentious as different studies have recognized the role of the environment and animals in the development, spread and spillover of antimicrobial resistant pathogens (ARPs) [8, 10]. On the other hand, studies have also reported that this relationship could be ambidirectional suggesting that humans could also spread the ESBL-Ec to animals and the environment [11, 12]. In addition, the inter-species transmission has also been reported globally (i.e., human to humans and animal to animal transmission) [13]. Indeed, there is a lot of undetected community carriage and transmission of ESBL-Ec at the human-animal-environment interface [14] even though there is inadequate local evidence for this. Such insufficient evidence deters the formulation of appropriate policies, strategies, and regulations to for the control of AMR [15, 16].

Following the global call for management of AMR (WHO "tricycle protocol"), the government of Uganda developed the National Action Plan (NAP) for AMR to protect human, animal, and environmental health [17]. This plan identifies a critical information gap regarding AMR especially at the Human-animal-environment interface. In areas like Wakiso district, an

urban farming area with a high level of antimicrobial agent use, AMR is not uncommon [18]. Coupled with having many livestock and humans, it presents a high opportunity for human-animal interaction hence providing a good platform to study ESBL-Ec transmission dynamics. In response to the WHO "tricycle protocol" and the Uganda NAP, our study described the epidemiology of ESBL-Ec at the human-animal-environment interface in a local peri-urban farming community in Wakiso district, central Uganda.

## Methods

### Study design and area

We conducted a cross-sectional study within farming communities of Wakiso district (00 24N, 32 29E). Wakiso district is the most populated district in Uganda with an average household size of 5 persons [19]. It is also one of the districts with the highest livestock numbers in Uganda [19]. The district has 503,442 households, 36% of which are engaged in both crop and animal husbandry [20]. The district is estimated to have these species of livestock: cattle (114,769), goats (132,964), sheep (27,542), pigs (199,962), chicken (2,783,509), ducks (33,350) and turkeys (4,852) [21]. Out of the total land area of 280,772.3 hectares, approximately 97,166.3 hectares (34.61%) are being used for agricultural production. Such demographic dynamics create a high probability of human-animal interaction.

### Sample size determination

Using Steve Bennett's formula of cluster sample size calculation [22], 104 households were targeted as a primary unit of sampling. At each household, four human samples, 4 animal samples and 4 environmental samples were picked for uniformity making a total of 1248 samples altogether. After thorough quality checks, a total of 988 samples passed and these proceeded to the next level of laboratory analysis. Animal samples (fecal per rectal), human samples (urine and stool) and environmental samples (doorknobs, soil, animal feeding equipment and water) were picked from the same household. Each of the households was counted as a batch and all its samples (placed in small Ziploc bags) were kept in the same bigger plastic courier bag. If any of the samples there in i.e. water, urine, fecal were found uncapped, or spilt, the entire batch was canceled and the household data rejected. These rejected households were then replaced by another household in order to maintain our sample size. In addition, any samples found without matching data i.e. sample descriptions and linkage to household data, were rejected. Altogether, 260 samples did not pass these quality checks.

### Study population and sampling procedure

Following a multi-stage sampling technique, our secondary sampling units (human, animal, and environment) was reached. All random sampling phases were conducted using a freely online available random number generator. Lists of villages from respective sub counties and parishes were obtained from the district veterinary office. Briefly, Wakiso district was purposively selected due to its unique presentation (see section on "Study design and area"). A total of 8 Sub counties were purposively (livestock keeping) selected with the help of a veterinary officer and from these 50% were randomly selected. From each Sub County, 50% parishes were randomly selected while villages with animals, and have had reported history of diarrhea episodes among humans were purposively selected with the help of local chairpersons and village health teams (VHTs). The households involved in the study needed to possess more than four animal species and at least two members in the household. The household was our primary sampling unit whereas humans, animals and environment comprised of the secondary

sampling unit. In addition, the household also served as a unit of analysis for the associations among humans, animals, and environment as far as the epidemiology of ESBL-Ec was concerned.

## Sample and data collection

A total of 104 farming households were studied between March and July 2022. Briefly, household related data, human sample donor data, environmental inspection data, and animal characteristics data were collected at each household using a mobile based application (kobo Toolbox). Pre-testing of the study tools was done among households in sub counties that were not part of this main study but within Wakiso district. A pre-tested semi-structured questionnaire was administered to household members who had experience with the key activities around the home.

A minimum of one and maximum of two individuals in a selected household were requested to provide a fecal and urine sample. A paper towel provided by the study teams and a sterile stool container with a scoop were used by each participant. Mid-stream urine was also provided by the participants during sampling. Parents always supported their children to collect the samples after a thorough consent and assent process. Additionally, animal samples were picked per rectum from a minimum of 4 animals within the sampled household. Environmental samples including soil, water for domestic use, swabs from animal feeding equipment and doorknobs were also picked. The sample containers were labelled with the unique code that represented the household and the sample category. Within four hours of sample collection, samples were delivered to the Microbiology Laboratory at the College of Veterinary Medicine Animal Resources and Biosecurity, Makerere University, in Ziploc bags under ice.

## Laboratory procedures

Samples were processed as soon as they arrived in the laboratory within 4 hours of collection. We ensured this by collecting fewer samples that could easily be processed upon arrival in the laboratory. Sample processing was done by aseptically transferring sample inoculum on to freshly prepared ESBL chromogenic agar (Condalab 2062, Madrid Spain) which is used for detection of gram-negative bacteria producing extended spectrum Beta lactamase. The original samples after culture were then kept in a fridge at 4°C for reference in case any of the cultures was not successfully done.

## Agar preparation and plating

Freshly prepared ESBL chromogenic agar containing ESBL supplement (condalab6042, Madrid Spain) having inhibition and selective agents was used [23]. Water and urine samples were first concentrated by centrifuging at 3000 rpm for 5 minutes prior to being introduced onto the agar. Soil samples and stool from Shoats were first emulsified/ suspended in 9mls of peptone water prior to plating. The plates were then incubated at 37°C for 24 hours at ambient humidity and air conditions. Presumptive ESBL-Ec appeared as pink, medium sized, raised, and moist colonies (Fig 1).

## Identification and confirmation

*E. coli* isolates were further confirmed by biochemical tests following manufacturer's instructions [24]. *E. coli* was confirmed in case the isolate was positive for indole production, methyl red, motile, and negative for citrate utilization, urease production, and Voges-Proskauer. Those isolates with reduced susceptibility to cefotaxime ($\leq$ 27 mm) and ceftazidime ($\leq$ 22

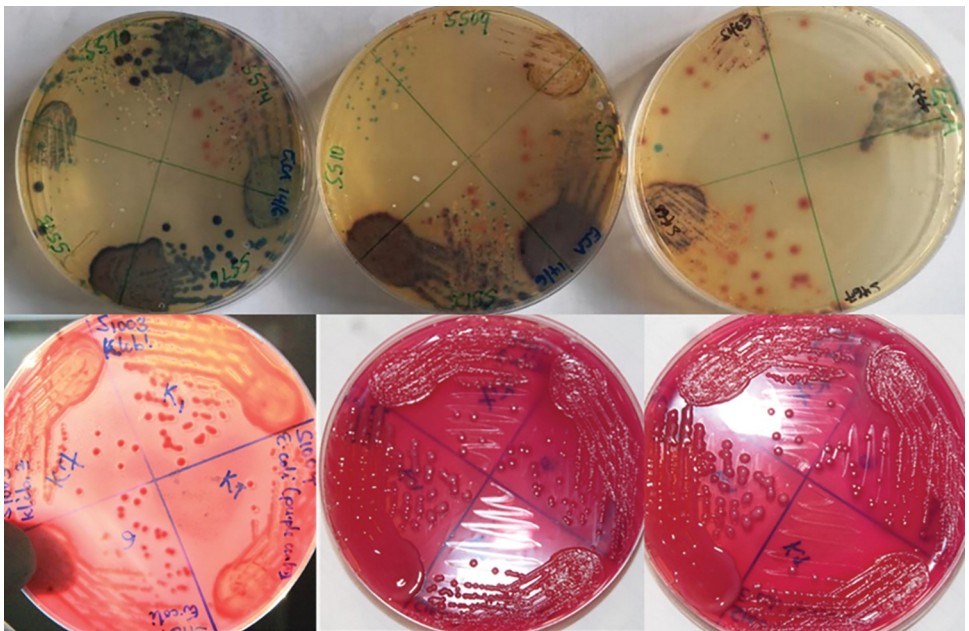

**Fig 1. Primary cultures, upper row and sub-cultures, lower row.**

mm) were confirmed for ESBLs- production using the modified double disk synergy (MDDS) method [25]. Briefly, after inoculation of the suspension onto Muller-Hinton agar (MHA), a disk of amoxicillin + clavulanic acid (20/10 μg) was placed in the center of the plate and then the disks of cefotaxime (25 μg) and ceftazidime (30 μg) were placed at 20 mm from the central disk on the same plate. The plates were then incubated at 35 ˚C for 24 hours and examined for an enhancement of inhibition zone of the β-lactam drugs caused by the synergy of the drugs and was interpreted as either being positive or negative for ESBLs-production (Fig 2).

## Study variables

Key independent variables for which data collected included individual demographics, household practices, animal husbandry practices, water, sanitation, and hygiene among other risk factors for occurrence of ESBL-Ec in communities. The dependent variable was the ESBL-Ec contamination status (positive or negative) of the household as defined by the status of the respective humans (carriage), animals, and the environment. The secondary dependent variable was ESBL-Ec human carriage which was defined as the presence of a positive sample i.e. either urine, or fecal or both in an individual.

## Data analysis and management

All data was cleaned for any missing variables or laboratory result to have a complete dataset. A new column was added for the outcome variable (laboratory outcome for each sample i.e., positive, or negative for ESBL-Ec). This procedure was conducted for all data from the humans, animals, and those from the environment therefore this study had four datasets. All data analysis procedures were done in R version 4.2.1. Selected variables were picked from each individual dataset (human, animal, and environment) and merged to the main household dataset. Summary statistics were run, percentages and frequencies were reported in tables. Overall prevalence was defined as the total number of positive samples out of the total number

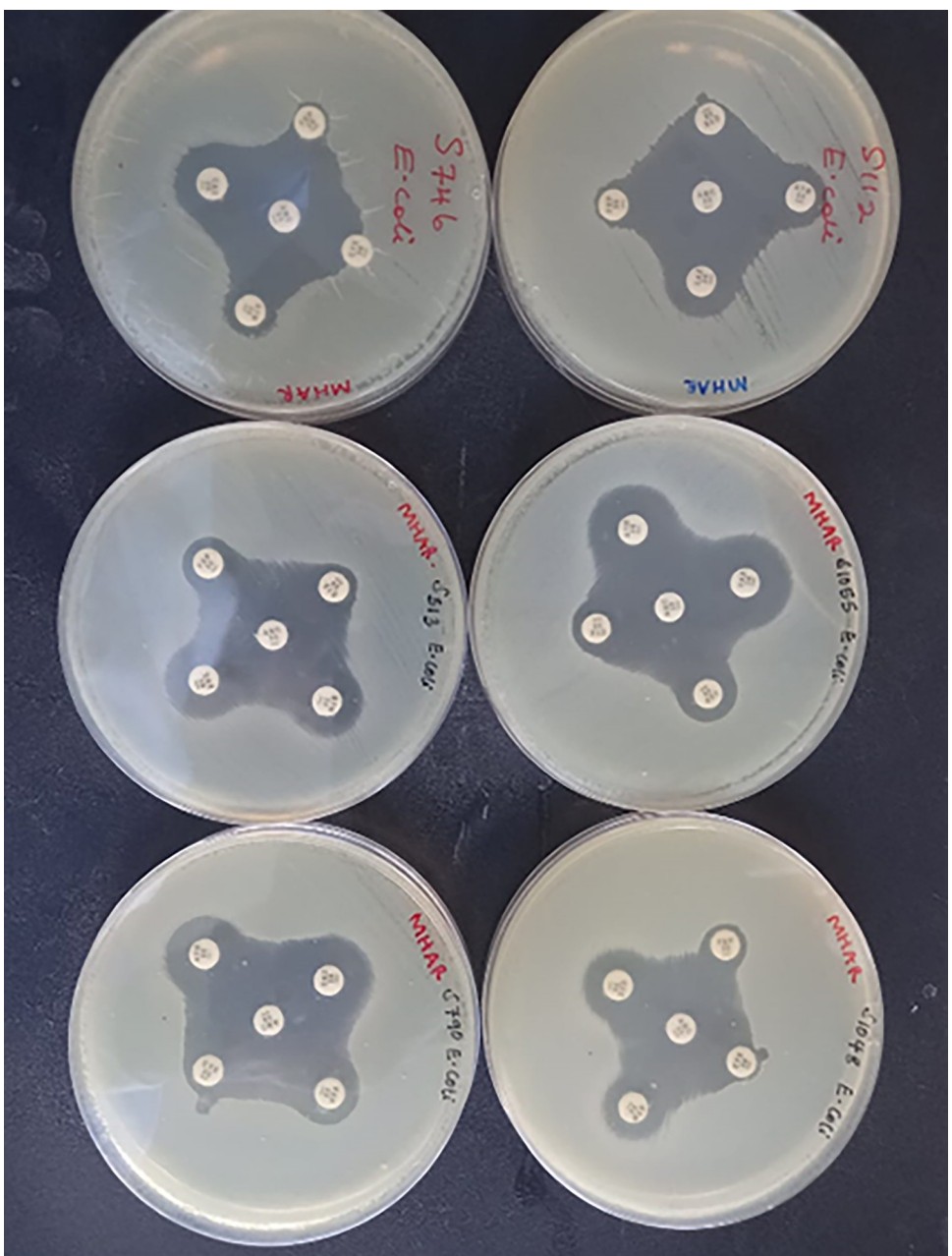

**Fig 2. Modified double disc diffusion test.**

of samples under consideration. Household prevalence on the other hand meant the total number of households with a positive ESBL-Ec sample out of the total number of households sampled. Bivariate and multivariate robust modified Poisson regression were performed to generate prevalence ratios (PR) and respective 95% confidence intervals.

## Quality control and assurance

Trained research assistants collected all environmental samples and corresponding data. Human participants were thoroughly trained on fecal and urine sample collection before

being given the sample collection containers. Upon provision of the sample from the human subject, the research assistants labelled and transported the samples while maintaining cold chain conditions. Trained veterinarians collected all animal samples during the study. All samples were verified to confirm the labeling, type, adequate volume, and integrity at the laboratory. Media quality control was also performed by inoculating standard bacterial strain of *E. coli* ATCC 25922 which is ESBL negative and *Klebshiella Pneumoniae* ATCC 700603 which is ESBL positive. The media quality control form was scored PASS in case the *E. coli* failed to grow and *K. pneumoniae* grew.

### Ethical considerations

The study sought ethical approval from Makerere University School of Public Health Higher Degrees Research and Ethics Committee (SPH-2021-167) and the Uganda National Council for Science and Technology (HS1919ES). We sought permission to conduct the study from Wakiso district headquarters (Chief administrative officer and district health and veterinary offices). Assent and consent forms were issued to the participants prior to their involvement into the study. Formal written consent was obtained from the parent/guardian before any child would be involved in the study. In addition, each of the abled children provided assent after being explained to the purpose of the study and its related processes. Personal identification information such as names, phone numbers were not collected. We however used unique codes for households based on the Sub County and village. All ethical issues, and confidentiality were followed as guided by the Helsinki declaration.

## Results

### Characteristics of the households involved in the study

This study was conducted among 104 households from four sub counties in Wakiso district. More than 60% (65/104) of the households had 5–9 members. Similarly, more than half of the sampled households had household heads who had attained primary school as the highest level of education, and more than 70% (74/104) of the total households were male-headed. In addition, 80% (83/104) of the households had a shared water source between humans and animals (Table 1).

### Characteristics of human and animal participants in the study

A total of 196 participants were involved in this study. About 50% (97/196) of the study participants were aged below 18 years whereas just 10% (19/196) were aged between 18–35 years. Over 57% (113/196) of the participants were females. Additionally, 31.6% (62/196) of the participants were farmers. About 32% (63/196) and 20.9% (41/196) of the participants had previously experienced gastrointestinal and urinal related illnesses respectively. Samples were collected from a total of 393 animals in our study. Compared to other animals, samples were mostly collected from shoats (sheep and goats) 25.0% (99/393). Intensive animal husbandry was the predominant form of animal husbandry, 37.0% (145/393). More than 80% (321/393) of the animals were reported to be treated by a veterinarian (Table 2).

### Burden (prevalence and severity) of ESBL-Ec among households in Wakiso district

More than 80.0% (86/104) of the households had at least one positive sample. Approximately 62.0% (64/104) of the sampled households were found to have humans carrying ESBL-Ec. Over 70.0% (77/104) of the sampled households had animals carrying ESBL-Ec. When assessed

**Table 1. Characteristics of the households studied.**

| Factors | Level | Frequency (n = 104) | Percentage (%) |
|---|---|---|---|
| Sub county of Wakiso district | Kakiri | 35 | 33.7 |
| | Kasangati | 25 | 24.0 |
| | Kasanje | 27 | 26.0 |
| | Kyengera | 17 | 16.4 |
| Household size | Below 5 | 21 | 20.2 |
| | 5–9 | 65 | 62.5 |
| Sex of head | Female | 30 | 28.9 |
| | Male | 74 | 71.1 |
| Highest education level of household head | Primary and below | 57 | 54.8 |
| | Secondary | 26 | 25.0 |
| | Tertiary and above | 21 | 20.2 |
| Form of location of the household | Rural | 54 | 52.0 |
| | Urban | 50 | 48.0 |
| Main water source for the household | Open | 23 | 22.1 |
| | Protected | 81 | 77.9 |
| Commonly shared water source between humans and animals | No | 21 | 20.2 |
| | Yes | 83 | 79.8 |
| Availability of hand washing facility | No | 60 | 57.7 |
| | Yes | 44 | 42.3 |
| Use animal waste in gardens | No | 15 | 14.4 |
| | Yes | 89 | 85.6 |
| Use human fecal in gardens | No | 90 | 86.5 |
| | Yes | 14 | 13.5 |
| Use human drugs to treat animals | No | 81 | 77.9 |
| | Yes | 23 | 22.1 |
| Availability of waste dumpsite | No | 65 | 62.5 |
| | Yes | 39 | 37.5 |
| Drinking water container has lid | No | 45 | 43.3 |
| | Yes | 59 | 56.7 |

at sample level, the overall prevalence of ESBL-Ec at the human-animal-environment interface among samples in Wakiso district was approximately 25.0% (251/988). At individual level, the humans, animals, and environment had ESBL-Ec prevalence of 35.4%, 55.4%, and 9.2% respectively. Approximately 17.0% (18/104) of the sampled households had no positive ESBL-Ec samples and were therefore considered to be low risk. A similar number of households however presented with a high burden implying that all the three sampled components were found to be positive for ESBL-Ec. Approximately 26.0% (27/104) of the households sampled had one of the components being positive (Table 3, Fig 3).

## Distribution of ESBL-Ec positivity among households

Fig 3 shows the different counts of households with the respective components. Majority of the positive households 45.3% (39/86) had both human and animal samples positive but not environment samples. Scenarios with only one component being positive yielded fewer representative households in the order (Animal>Human>Environment). A total of 17 households had their human, animal and environment samples positive for ESBL-Ec (Fig 3).

**Table 2. Characteristics of human participants and animal subjects.**

| Factors | Level | Frequency (n = 196) | Percentage (%) |
|---|---|---|---|
| **Human participants** | | | |
| Age | Less than 18 | 97 | 49.5 |
| | 18–35 | 19 | 9.7 |
| | Over 35 | 80 | 40.8 |
| Sex | Female | 113 | 57.7 |
| | Male | 83 | 42.3 |
| Education level | Formal | 160 | 81.6 |
| | None | 36 | 18.4 |
| Occupation | Farmer | 62 | 31.6 |
| | None | 28 | 14.3 |
| | Others | 26 | 13.3 |
| | Student | 80 | 40.8 |
| Common condition you suffer from | Intestinal | 63 | 32.1 |
| | Urinal | 41 | 20.9 |
| | None | 92 | 46.9 |
| Sometimes did not complete treatment | No | 98 | 50.0 |
| | Yes | 98 | 50.0 |
| Often closely interacted with animals | No | 32 | 16.5 |
| | Yes | 162 | 83.5 |
| **Animal subjects (n = 393)** | | | |
| Species | [a]Shoats | 99 | 25.2 |
| | [b]Poultry | 92 | 23.4 |
| | [c]Companion animals | 66 | 16.8 |
| | Pig | 78 | 19.8 |
| | Cattle | 58 | 14.8 |
| Husbandry practice | Backyard | 110 | 28.0 |
| | Free range | 138 | 35.1 |
| | Intensive | 145 | 36.9 |
| Feeding strategy | Kitchen leftover | 89 | 22.7 |
| | Natural feeds (grass) | 254 | 64.6 |
| | Supplementary feeds (mixed) | 50 | 12.7 |
| Source of water for animals | Protected | 276 | 70.2 |
| | Unprotected | 117 | 29.8 |
| Treatment options | Experienced farmer | 5 | 1.3 |
| | Owner treated | 67 | 17.1 |
| | Veterinarian | 321 | 81.7 |

[a]Shoats

[b]Poultry meant turkey, chicken, and ducks

[c]Companion animals included: rabbits, cats and dogs

## Factors associated with ESBL-Ec human carriage and household contamination

Several factors (recent visitors, source of water, use of protective lid for water, etc) were associated with ESBL-Ec human carriage among the study population. Notably, households that had received a visitor in the previous 24 hours before sample collection had a higher human ESBL PE carriage (adj PR = 1.34, 95% CI: 1.13–1.60) compared to those that didn't have a visitor.

**Table 3. Burden of ESBL-Ec among households in Wakiso district.**

| Component | Level | Frequency | Percentage (95% CI) |
|---|---|---|---|
| **Prevalence of ESBL-Ec by household** | | | |
| Overall household (N = 104) | | 86 | 82.7 (73.8–89.2) |
| Humans | | 64 | 61.5 (51.5–70.8) |
| | Urine | 29 | 27.9 (19.8–37.7) |
| | Fecal | 55 | 52.9 (42.9–62.7) |
| Animals | | 77 | 74.0 (64.4–81.9) |
| Environment | | 21 | 20.2 (13.2–29.4) |
| **Prevalence of ESBL-Ec by Individual samples** | | | |
| Overall | | 251 | 25.4 (22.7–28.3) |
| Human | | 89 | 35.4 (30.7–40.3) |
| Animals | | 139 | 55.4(50.0–61.6) |
| Environment | | 23 | 9.2(6.0–13.6) |
| **Severity of the ESBL-Ec infection among households** | | | |
| No threat | | 18 | 17.3 (10.8–26.2) |
| Low burden | | 27 | 26.0 (18.1–35.6) |
| Medium burden | | 41 | 39.4(30.1–49.5) |
| High burden | | 18 | 17.3(10.8–26.2) |

The prevalence of ESBL-Ec was lower among those that collected water from protected water sources such as taps, protected springs compared to those that collected water from unprotected water sources such as wells (adj PR = 0.63, 95%CI: 0.46–0.88). Participants in households where the drinking water was stored from a container with a lid had a lower prevalence of ESBL-Ec (adj PR = 0.84 95% CI: 0.73–0.96), in addition, households whose environment was found with dirt also had a lower burden of ESBL-Ec among humans (adj PR = 0.71 95% CI: 0.53–0.96) compared to the households with clean environments. We found out that households that had a visitor in the previous 24 hours preceding sample collection were associated with high ESBL-Ec household contamination (adj PR = 1.19, 95% CI: 1.04–1.36) compared to those that had no visitors. Notably, households that used veterinary workers to

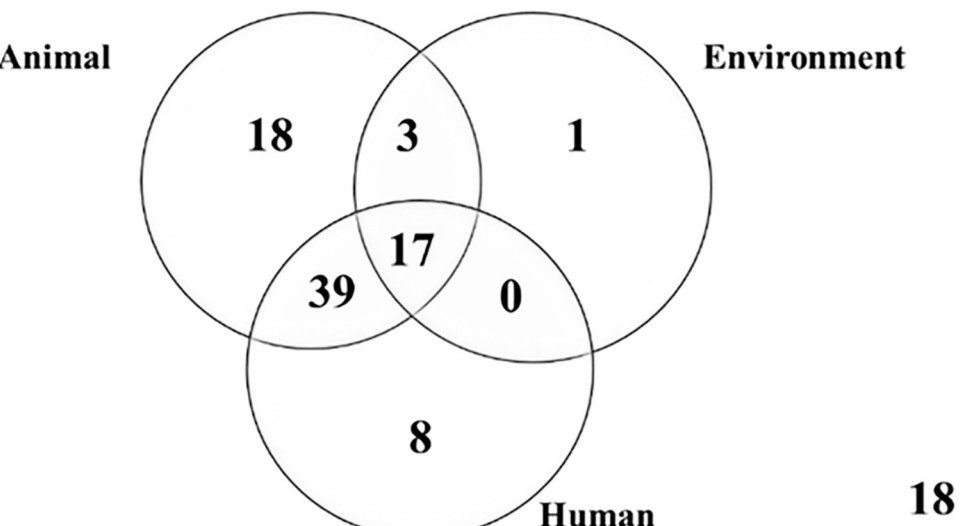

**Fig 3. Household sharing of ESBL-Ec bacteria.**

**Table 4. Factors associated with ESBL-Ec human carriage and household contamination.**

| Variable | Level | Un adjusted PR (95% CI) | adjusted PR (95% CI) | p-value |
|---|---|---|---|---|
| **Factors associated with human ESBL-Ec carriage** | | | | |
| Household that had a visitor in the last 24 hours | No | 1 | Ref | |
| | Yes | 0.92(0.81–1.03) | 1.34(1.13–1.60) | 0.0009*** |
| Protected Water source used as the main water source | No | 1 | Ref | |
| | Yes | **0.85(0.76–0.95) | 0.63(0.46–0.88) | 0.006** |
| Presence of a hand washing facility at home | No | 1 | Ref | |
| | Yes | *0.86(0.76–0.97) | 1.21(0.91–1.62) | 0.19 |
| Drugs administered to sick animals by a veterinary-personnel | No | 1 | Ref | |
| | Yes | **1.39(1.13–1.71) | 1.10(0.85–1.39) | 0.48 |
| Guidance on how long to wait after treating animal provided by veterinary personnel | No | 1 | Ref | |
| | Yes | 1.08(0.96–1.23) | 0.84(0.70–1.01) | 0.068 |
| Household compound found dirty | No | 1 | Ref | |
| | Yes | 0.93(0.83–1.05) | 0.71(0.53–0.96) | 0.024* |
| Presence of stagnant water | No | 1 | Ref | |
| | Yes | *0.88(0.76–1.00) | 0.93(0.70–1.23) | 0.59 |
| Drinking water container with lid | No | 1 | Ref | |
| | Yes | *0.87(0.78–0.98) | 0.84(0.73–0.96) | 0.014* |
| **Factors associated with ESBL-Ec household contamination** | | | | |
| Household had a visitor in the last 24 hours | No | 1 | 1 | Ref |
| | Yes | 0.95(0.88–1.04) | 1.19(1.04–1.36) | 0.009** |
| Presence of household member who had been to another district or country in the past one month | No | 1 | 1 | Ref |
| | Yes | *1.08(1.01–1.16) | 1.11(0.94–1.30) | 0.23 |
| Source of drugs | Nonprofessional | 1 | 1 | Ref |
| | Professional | *1.15(0.99–1.33) | 1.39(1.20–1.61) | <0.05*** |
| Reasons for negotiating on treatment | Financial | 1 | 1 | Ref |
| | Side effects | 1.09(0.97–1.23) | 1.09(1.00–1.19) | 0.06. |
| Cleaning of animal house | Twice a week | 1 | 1 | Ref |
| | Once a week | *1.10(1.00–1.19) | 1.06(0.95–1.18) | 0.25 |
| | Throughout a week | 0.97(0.87–1.07) | 0.95(0.84–1.07) | 0.41 |
| Management of animal waste | Dispose | 1 | 1 | Ref |
| | Use | 1.11(0.96–1.29) | 1.29(1.05–1.60) | 0.02* |
| Drugs administered to sick animals by veterinary personnel | No | 1 | 1 | Ref |
| | Yes | *1.28(1.05–1.58) | 1.39(1.20–1.61) | <0.05** |

administer drugs to their sick animals were significantly associated with a higher ESBL-Ec contamination in households (adj PR = 1.39, 95% CI: 1.20–1.61) and, those that used animal waste were more likely to be contaminated with ESBL-Ec (adj PR = 1.29, 95% CI: 1.05–1.60). Additionally, households that reported getting medicines from health workers were more likely to be contaminated with ESBL-Ec (adj PR = 1.39, 95% CI: 1.20–1.61) compared to those that reported engaging in other sources such as friends, self-medication, relatives among others (Table 4).

## Discussion

The burden of ESBL-Ec carriage among apparently healthy individuals and animals has been reported to be high globally with limited evidence on the environment [26]. There is scanty

information about the ESBL-Ec among humans, animals and their immediate environment especially in LMICs such as Uganda. Surveillance of ESBL-Ec "globally recognized sentinel organism for AMR" at the human-animal-environment interface provides information required to design sustainable one health interventions globally. In our study, we report a high household ESBL-Ec burden. Our findings showed that two factors: visit by animal health workers and visitation to a human health facility were highly associated with human ESBL-Ec carriage and household contamination. These two results are indicative of the poor infection prevention and control measures by both the human and animal health workers thus presenting a great concern for public health.

This study reports an overall ESBL-Ec prevalence of 25.4% from all the components sampled (i.e. humans, animals and environment). On the contrary, authors in Nepal reported 53.4% prevalence of ESBL-Ec at the human-animal-environment interface [10]. This difference could potentially be attributed to the variation in environmental sampling in the two studies. The earlier study considered community drainage and sewage systems around households which overestimated the ESBL-Ec prevalence at household level while this study utilized the immediate household environment that yielded a comparatively low ESBL-Ec. Over 83% of the sampled households had a positive ESBL-Ec sample irrespective of its source. Such findings reveal a high level of household contamination and risk of ESBL-Ec infections especially in a typical farming community. In case of an infection and disease, this situation will certainly drive the cost of treatment high, long stay in the hospitals and could compromise quality of health outcomes for both humans and animals [27].

Majority of the households had animal samples contributing highest to ESBL-Ec household positivity followed by human samples. This result could be attributed to the fact that *E. coli* is a harmless inhabitant of the human and animal guts [28] apart from its key role in the transmission of ESBL genes. The regular and irrational use of antimicrobials in animal production could also trigger the development of ESBL-Ec in animals [29]. Several human related factors such as self-medication, over the counter medicines, animal food products containing drug residues and the general transmission dynamics of the organisms could explain the observed burden among humans [8]. Perhaps the low contribution to environment ESBL-Ec prevalence is because the environment subjects the bacteria to harsh conditions and therefore their population is constantly being checked [30]. Studies that have documented a high burden of ESBL-Ec in the environment have sampled the sites with constant supply of nutrients for the survival of the organisms such as storm water drains and sewerage systems [31]. Our study utilized the immediate environment of humans and animals to get a clear indicator of transmission dynamics of ESBL-Ec at household level. It is therefore important that future studies focus on the survival and viability of ESBL-Ec in environmental components exposed to harsh weather conditions such as sunshine and heavy rainfall. Households that had a visitor in the previous 24 hours before sample collection had a higher ESBL-Ec contamination compared to those that never had a visitor. Interestingly, also humans from households that had a visitor 24 hours preceding sample collection registered a high ESBL-Ec carriage. Indeed, visitors have been earlier implicated in not only ESBL-Ec occurrence but also the general AMR transmission cycle [32, 33]. This could potentially be attributed to the fact that visitors come from different locations, use different transport means and might be carrying resistant organisms such as ESBL-Ec. Infection prevention and control strategies such as hand washing with soap and sanitizing should be emphasized at household level at all critical hand washing times including when an individual just arrives home as has been promoted in the control of the spread of the COVID-19 pandemic.

Important to note, obtaining water from a protected source was associated with a low ESBL-Ec carriage among humans. Protected water sources provide wholesome water free

form contamination by humans, animals, and storm water [34]. Unprotected water sources are vulnerable to a wide range of contamination including ESBL-Ec from the environment and anthropogenic activities [35]. In line with the safe water chain, our study revealed that having drinking water containers covered with lids was associated to low ESBL-Ec human carriage. Safe water storage reduces the bacterial load and further contamination is prevented by covering the storage container with a lid [36]. This is similar to a finding by Brick et al who reported that covering drinking water container with a lid limited contamination [37]. Therefore, there is need for communities to ensure a safe water chain from the water source up to use.

Interestingly, humans from households with compounds having dirt were found to have less ESBL-Ec carriage. A study conducted on improving surface sampling and detection of contamination presents different views and contentions around dirty environments and safety [38]. This could be due to several factors such as use of personal protective equipment (PPE), low survival of ESBL-Ec in the harsh environment, proper personal hygiene practices, among others in farming communities. Several studies have indicated that good hygienic practices, increased household use of PPE and low abundance of bacterial pathogens in the environment could potentially lead to reduced contamination especially with ESBL-Ec organisms [30, 39]. Therefore, more environmental studies are required on bacterial abundance and AMR especially in household environments.

The households that got the drugs from the professional healthcare providers were more likely to be contaminated with ESBL-Ec. Given the high burden of drug resistant organisms, household members who attend or seek medical attention from health care facilities were at higher risk of contracting ESBL-Ec and other nosocomial infections [40]. Humans have ability to shed the contracted bacteria into the environment and or with the animals with in the household setting [8]. Therefore, it is important to improve the infection prevention and control in the health facilities to minimize hospital acquired infections among community members.

Households that contacted a veterinary worker to administer the drugs to their sick animal were more likely to be at risk of ESBL-Ec contamination. Poor on farm biosafety measures, improper compliance to the use of PPE, reuse of veterinary equipment among farms, poor storage of veterinary drugs among others could potentially contribute to the reported household contamination by veterinary workers [41]. In addition, misuse of veterinary products without prior diagnosis could escalate resistance among animals in households [42]. Therefore, improved farm biosafety measures, good veterinary practices as well as veterinary drug use regulations are paramount to reverse the problem of AMR.

Households that used the animal waste in several activities such as gardening, biogas production were more likely to be contaminated with ESBL-Ec than those that disposed of the waste. Studies have indicated the presence of ESBL-Ec in animal waste [39]. Utilization of such waste without decreasing the contamination load and without biosafety precautions exposes the human, other animals and the environment to subsequent ESBL-Ec contamination [43]. Therefore, the use of PPE during waste handling, decontamination of animal waste before use should be emphasized among farming households in order to curb contamination and ESBL-Ec transmission. Our study assessed practices linked to spread and spillover of infectious organisms in a farming household. However, as a limitation, these practices were reported instead of being observed. This limitation was overcome by first creating a good rapport with the respondents to ensure that the responses obtained on practices were as close as possible to the actual practice if it were to be observed. Our study did not characterize the ESBL-Ec organisms genotypically even though a phenotypic analysis was made. Future studies should benefit more from a genotypic characterization of these ESBL-Ec organisms especially at the one health interface.

## Conclusion

This study is among the first of its kind in Uganda focusing at determining the ESBL-Ec carriage, contamination, and associated risk factors at household level among farming communities. A wide spread of ESBL-Ec among humans, environment and animals indicates a great public health threat. This observation could potentially be due to poor infection prevention and control (IPC) measures in the area by veterinarians and household members. The role of human, environment, and animal health workers in the occurrence of ESBL-Ec and other types of AMR is therefore critical.

## Recommendations

Improved collaborative AMR mitigation strategies such as safe water chain, on farm biosecurity, household, and facility-based IPC measures as well as capacity building of the human, animal and environmental health workers using a one health approach is paramount in order to curb the problem of AMR among farming communities. Veterinarians should be trained in IPC measures and encouraged to disinfect their equipment and sanitize their hands between farms. Further studies in agricultural farmlands need to be done to check for the presence of the beta-lactam drug residues and or actual presence of the ESBL-producing organisms that perhaps could be contaminating the household environments.

## Supporting information

**S1 File. Letters of support and study approval.**
(PDF)

**S2 File. Evidence for PhD study.**
(PDF)

**S1 Data. Dataset.**
(XLS)

## Acknowledgments

We are grateful to the respondents who participated in this study. We also thank the district and sub county officials who supported the study inception and processes. The local authorities in the respective study sites were very supportive and our research participants in the same regard.

## Author Contributions

**Conceptualization:** James Muleme, David Musoke, Stevens Kisaka, Frederick E. Makumbi, Esther Buregyeya, John Bosco Isunju, Rogers Wambi, Richard K. Mugambe, Clovice Kankya, Musso Munyeme, John C. Ssempebwa.

**Data curation:** James Muleme, Bonny E. Balugaba, Stevens Kisaka, Frederick E. Makumbi.

**Formal analysis:** James Muleme, Bonny E. Balugaba.

**Funding acquisition:** John Bosco Isunju, Clovice Kankya, John C. Ssempebwa.

**Investigation:** James Muleme, Esther Buregyeya, John Bosco Isunju, Rogers Wambi, Richard K. Mugambe, John C. Ssempebwa.

**Methodology:** James Muleme, David Musoke, Bonny E. Balugaba, Frederick E. Makumbi, Esther Buregyeya, John Bosco Isunju, Richard K. Mugambe, Clovice Kankya, Musso Munyeme, John C. Ssempebwa.

**Project administration:** James Muleme, Rogers Wambi.

**Resources:** James Muleme, Esther Buregyeya, John Bosco Isunju, Clovice Kankya, John C. Ssempebwa.

**Software:** Bonny E. Balugaba.

**Supervision:** James Muleme, David Musoke, Stevens Kisaka, Frederick E. Makumbi, Esther Buregyeya, Richard K. Mugambe, Clovice Kankya, Musso Munyeme, John C. Ssempebwa.

**Validation:** James Muleme, Bonny E. Balugaba, Stevens Kisaka, Frederick E. Makumbi, Esther Buregyeya, Richard K. Mugambe, Clovice Kankya, Musso Munyeme, John C. Ssempebwa.

**Visualization:** James Muleme, Bonny E. Balugaba, Stevens Kisaka, Rogers Wambi, John C. Ssempebwa.

**Writing – original draft:** James Muleme, Bonny E. Balugaba, Stevens Kisaka, John Bosco Isunju, Rogers Wambi, Musso Munyeme, John C. Ssempebwa.

**Writing – review & editing:** James Muleme, David Musoke, Bonny E. Balugaba, Stevens Kisaka, Frederick E. Makumbi, Esther Buregyeya, John Bosco Isunju, Rogers Wambi, Richard K. Mugambe, Clovice Kankya, Musso Munyeme, John C. Ssempebwa.

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
