## [Decision Letter · Decision Letter 0]

5 Jan 2023

PGPH-D-22-01776

Epidemiology of Extended-spectrum beta-lactamase-producing Escherichia coli at the human-animal-environment interface in Wakiso district, Uganda

Dear Dr. Muleme,

Thank you for submitting your manuscript to PLOS Global Public Health. After careful consideration, we feel that it has merit but does not fully meet PLOS Global Public Health’s publication criteria as it currently stands. Therefore, we invite you to submit a revised version of the manuscript that addresses the points raised during the review process.

We look forward to receiving your revised manuscript.

Kind regards,

Ismail Ayoade Odetokun, DVM, Ph.D.

Academic Editor

Journal Requirements:

2. Please provide separate figure files in .tif or .eps format only and remove any figures embedded in your manuscript file. Please also ensure that all files are under our size limit of 10MB.

3. Your manuscript is missing the following sections: Introduction. Please ensure these are present, and in the correct order, and that any references to subheadings in your main text are correct. An outline of the required sections can be consulted in our submission guidelines here:

https://journals.plos.org/globalpublichealth/s/submission-guidelines#loc-parts-of-a-submission

4. We have noticed that you have uploaded Supporting Information files, but you have not included a list of legends. Please add a full list of legends for your Supporting Information files after the references list. 

5. In the online submission form, you indicated that "Data will be made available upon request". All PLOS journals now require all data underlying the findings described in their manuscript to be freely available to other researchers, either 1. In a public repository, 2. Within the manuscript itself, or 3. Uploaded as supplementary information.

Additional Editor Comments (if provided):

Reviewers' comments:

Reviewer's Responses to Questions

**Comments to the Author**

1. Does this manuscript meet PLOS Global Public Health’s publication criteria? Is the manuscript technically sound, and do the data support the conclusions? The manuscript must describe methodologically and ethically rigorous research with conclusions that are appropriately drawn based on the data presented.

Reviewer #1: Yes

Reviewer #2: Yes

Reviewer #3: Partly

2. Has the statistical analysis been performed appropriately and rigorously?

Reviewer #1: Yes

Reviewer #2: Yes

Reviewer #3: Yes

3. Have the authors made all data underlying the findings in their manuscript fully available (please refer to the Data Availability Statement at the start of the manuscript PDF file)?

Reviewer #1: Yes

Reviewer #2: Yes

Reviewer #3: No

4. Is the manuscript presented in an intelligible fashion and written in standard English?

Reviewer #1: Yes

Reviewer #2: No

Reviewer #3: Yes

5. Review Comments to the Author

Reviewer #1: 1-Recent references are recommended.

2-Why authors did not use PCR as confirmatory tool for detection of ESBL genes???, phenotypic detection of ESBL strains is not enough.

3-Separation between conclusion and recommendations is necessary.

4-Clear strategies in order to protect human, animal and environment from ESBL are needed.

Reviewer #2: The significance of the One Health concept in the fight against AMR cannot be overstated, thus the significance of this study. It is suggested that authors review the following as stated in the comments:

Line 124- What quality standards were utilized to eliminate obtained samples?

Line 129- Address the multistage sampling strategy referred to as "figure 1" on line 129 is nowhere further included in the text. While "figure 1" on line 170 depicts primary and sub-cultures.

The entire Table and figure labeling should be addressed and corrected. (Lines 264, 265 & 296)

Line 149 & 205- The process of collecting samples needs additional elaboration and explanation. There was a discrepancy between the statement in line 205 that research assistants are responsible for sample collection after receiving proper training and the instruction provided to study participants regarding collection procedures in line 149.

These require clarification for proper comprehension.

Line 160-How were the samples stored before they were processed in the laboratory, and was the processing done as soon as the samples arrived in the laboratory?

Line 242- Were some practice questions administered, observed, or reported by the participants? If they were reported, those should be stated and possibly signified as a limitation of the study since participants were not observed carrying out those practices.

Lines 241 - 242 & 259- Error referencing in results should be clarified.

Lines 271- 273- The sentence should be restructured for ease of understanding and clarity.

Line 296- All tables should be re-formatted, and authors should consider reporting only adjusted PR instead of having unadjusted PR in the table.

To correct all grammatical and spelling errors, the entire manuscript should be reviewed by a native English speaker.

Reviewer #3: Muleme et al., studied the Epidemiology of Extended-spectrum beta-lactamase-producing Escherichia coli at the human-animal-environment interface in Wakiso district, Uganda.

My comments are as follows:

Abstract

1. Line 38. Replace the word explained and write in past tense. Eg examined

2. Line 48. You cannot have a total of percentage. Rephrase as a total of ** households or write the percentage and remove the “a total of”

3. Line 53. You never mentioned the word “gardening” in the entire manuscript. Kindly check line 387.

Introduction

1. Reference 1 is on Uganda NPHC report. I DO NOT think that report is the correct reference for line 67-69.

2. Line 71. Reference 2 is missing

3. Line 72. Rephrase this statement as: Studies have reported that the resistance to third generation cephalosporins in E. coli ranged from 0–87% whereas resistance to fluoro……..

4. Line 74. Change and colleagues to “et al”

5. Co-carriage with what? Fluoroquinolones?

6. Materials and Methods

Materials and Methods

7. Line 103. Rephrase as: our study “described” and remove “was formulated aiming at”.

8. Line 110. Include “average” before household and include unit such as 4.7 persons.

9. Line 118-127. Could you justify why 260 samples (1248-988) were sub-optimal and did not pass your “thorough quality check”?

10. If your formulae gave you 840 samples, why did you over-sample and collected 1248 out of which 260 were bad and you ended up with 988?

11. Line 129. Change secondary samples to secondary sampling units.

12. Line 143. Is the word recruited appropriate for households?

13. Kindly state that no personal identifying information such as name, phone number or other materials were collected. If they were collected, how did you ensure their protection?

14. Line162. Include the country of the Chromoger.

15. Line 182. Most labs would culture MHA meant for AST at 35 degree and not 37. Kindly confirm this.

Results

16. Line 226. Add “and” before 71.1%

17. Table 1. Factor number 11 and 13 do not add up to 104. Cross-check these.

18. Line 232. How did you arrive at 196 participants? One would assume that 4 individual per household (n=104) would result in 416 human participants “(samples).

19. Line 232. Please remove were recruited to provide samples.

20. Line 236. It would be difficult to separate stomach from intestinal. SO, I would suggest gastrointestinal illnesses rather than intestinal.

21. Line 236. Change “had been commonly suffering” to “have previously experienced”

22. Line 238-239. Kindly include the % in the bracket before the fraction.

23. Line 241-242. Correct the error in the Table.

24. Line 243. Shoat does not mean sheep and goat. Kindly confirm the meeting and edit appropriately.

25. Line 253. Change “up to” to “approximately”

26. Line 259. Revise the highlighted “Error”

27. Line 264. Include Figure number

28. Line 265. Delete Figure from title of table, Delete and households too in Title.

29. Line 277-278 and Line 286-287 appears as a repetition to me. Could you please differentiate these statements?

30. Line 290. Kindly report in past tense. Those that “used” not “using”

31. Line 296. Correct table title.

32. Line 296. Table seems unconvincing. A. Could you tell me what your outcome variable was? B. Was the LR model based on presence of ESBL-PE or absence of ESBL-PE?

C. Could you tell me why a household with dirty environment could have less prevalence than a clean household or why drugs administered by a Vet could have resulted in more ESBL than if it were administered by non-vets?

DISCUSSION

33. Line 316. Change round to around

34. Line 322. Change as well as to “and could compromise…”

35. Line 325. Include “the” before the least contributor.

36. Rephrase line 357. I suggest : “This is similar to the report of Brick et al who reported that …….

37. Your references are not formatted to the Vancouver Standard.

38. Your last reference was 30. However from Line 335, you have continued with 31,32 etc. So, update your reference list and format correctly.

39.

6. PLOS authors have the option to publish the peer review history of their article (what does this mean?). If published, this will include your full peer review and any attached files.

**Do you want your identity to be public for this peer review?** For information about this choice, including consent withdrawal, please see our Privacy Policy.

Reviewer #1: No

Reviewer #2: No

Reviewer #3: No

---

## [Decision Letter · Decision Letter 1]

23 Mar 2023

PGPH-D-22-01776R1

Epidemiology of Extended-spectrum beta-lactamase-producing Escherichia coli at the human-animal-environment interface in a farming community of central Uganda

Dear Dr. Muleme,

Thank you for submitting your manuscript to PLOS Global Public Health. After careful consideration, we feel that it has merit but does not fully meet PLOS Global Public Health’s publication criteria as it currently stands. Therefore, we invite you to submit a revised version of the manuscript that addresses the points raised during the review process.

We look forward to receiving your revised manuscript.

Kind regards,

Ismail Ayoade Odetokun, DVM, Ph.D.

Academic Editor

Journal Requirements:

Additional Editor Comments (if provided):

Reviewers' comments:

Reviewer's Responses to Questions

**Comments to the Author**

1. If the authors have adequately addressed your comments raised in a previous round of review and you feel that this manuscript is now acceptable for publication, you may indicate that here to bypass the “Comments to the Author” section, enter your conflict of interest statement in the “Confidential to Editor” section, and submit your "Accept" recommendation.

Reviewer #2: (No Response)

Reviewer #3: (No Response)

2. Does this manuscript meet PLOS Global Public Health’s publication criteria? Is the manuscript technically sound, and do the data support the conclusions? The manuscript must describe methodologically and ethically rigorous research with conclusions that are appropriately drawn based on the data presented.

Reviewer #2: Yes

Reviewer #3: Partly

3. Has the statistical analysis been performed appropriately and rigorously?

Reviewer #2: Yes

Reviewer #3: Yes

4. Have the authors made all data underlying the findings in their manuscript fully available (please refer to the Data Availability Statement at the start of the manuscript PDF file)?

Reviewer #2: Yes

Reviewer #3: Yes

5. Is the manuscript presented in an intelligible fashion and written in standard English?

Reviewer #2: Yes

Reviewer #3: Yes

6. Review Comments to the Author

Reviewer #2: We appreciate your efforts to correct all observed comments. However, kindly address the following observations as well.

Line 125- How was cross-contamination avoided if samples from humans, the environment, and animals were kept in the same plastic bag?

Line 154- Explain the process of selecting one individual in a selected household who was requested to provide a fecal and urine sample.

Results section- Tables need to be adequately labeled.

Reviewer #3: You have diligently reviewed the manuscript. Congratulations.

Could you look into the following.

Line 51. As much as possible, dont use the word ”However”. So, delete and start the statement with ””Covering the ....”.

Line 70. A review by ……… cannot have three references. I suggest you change to studies conducted in “xxx country” showed

Line 75. Relationships or interactions?

Line 78 & 79. Italicize E.coli

Line 82. Change since different studies to ….. as different studies have recognized the role of the environment ….. Kindly write in past tense as much as possible

Line 117. Am I right to assume that you wanted to sample 104 households, you got 12 samples from each (total of 1248 and 988 passed your sample screening test). Where does the 840 come from?

Line 118. Delete the “,” after (23)

Line 132. Change approach to technique

Line 149. Rephrase as : mobile based application (kobo Toolbox).

Line 176. The wrong word: incubated. Do you mean : freshly prepared?

Line 180, 193. Use o instead as the 0 is wrong

Line 185. Standard protocols or manufacturer instructions and not standard operating procedures.

Line 242. Remove the bracket on “5-9 members”

Line 154 said : one individual was requested to provide samples per household, Now line 249. You said 196 study participants, How did this come about? Kindly reconcile

Line 257-258, 280, 287 – Typographical error in Table title

Line 255 – You said shoats – 99/393 and in Table pig is 78… you need to reconcile this

Table ****…. ANIMAL SUBJECTS: I think the denominator should be the number of household and not number of animals. Do you think there would be difference in husbandary practice, feeding, source of water, and treatment option per animal belonging to the same household. I think it will make more sense if its based on household and not animals. Instead of 321 animals being treated by a Vet, you state **% of animal owners consulted a vet to treat their animals. What do you think?

Line 260-266. Is repetition of methods and no results therein. Consider deleting it or move to methods. Check Line 123 and 159-160.

Line 269. This was result of analysis of field samples. Why do you need a CI:. You don’t need CI for isolates retrieved from samples.

Line 291. AN opening statement would make it easier to read. Something like: several factors (recent visitors, source of water, use of protective lid for water, etc) were associated withESBL-PE human carriage among the study population.

Line 291-293 Contrdicts line 299-301. Why the results (OR and 95% CI) different?

Line 317. Please remove “the” before LMICs

Line 320. Rephrase the statement to begin as: “Our findings showed that two factors: visit by animal health workers and visitation to a human health facility were highly associated……. “

Line 353-356. Looks like repetition despite the differences: sample collection and data collection. That then raises the question why did you collect them on different days? I suggest you delete one to save you some stress.

Line 362. Am i correct that COVID19 or COVID-19 is the most common way of writing

Line 375. Since you mentioned the study... you should state the different views and contentions as argued by that study.

Line 387. Kindly rephrase as: ... were at higher risk of contracting ESBL-PE and other nosocomial infections (41).

Am happy to recommend the manuscript if these observations have been addressed.

Best wishes

7. PLOS authors have the option to publish the peer review history of their article (what does this mean?). If published, this will include your full peer review and any attached files.

**Do you want your identity to be public for this peer review?** For information about this choice, including consent withdrawal, please see our Privacy Policy.

Reviewer #2: No

Reviewer #3: No

---

## [Decision Letter · Decision Letter 2]

17 Apr 2023

PGPH-D-22-01776R2

Epidemiology of Extended-spectrum beta-lactamase-producing Escherichia coli at the human-animal-environment interface in a farming community of central Uganda

Dear Dr. Muleme,

Thank you for submitting your manuscript to PLOS Global Public Health. After careful consideration, we feel that it has merit but does not fully meet PLOS Global Public Health’s publication criteria as it currently stands. Therefore, we invite you to submit a revised version of the manuscript that addresses the points raised during the review process.

We look forward to receiving your revised manuscript.

Kind regards,

Ismail Ayoade Odetokun, DVM, Ph.D.

Academic Editor

Journal Requirements:

Additional Editor Comments (if provided):

Please pay attention to the minor comments from the reviewers.

Reviewers' comments:

Reviewer's Responses to Questions

**Comments to the Author**

1. If the authors have adequately addressed your comments raised in a previous round of review and you feel that this manuscript is now acceptable for publication, you may indicate that here to bypass the “Comments to the Author” section, enter your conflict of interest statement in the “Confidential to Editor” section, and submit your "Accept" recommendation.

Reviewer #2: (No Response)

Reviewer #3: All comments have been addressed

Reviewer #4: All comments have been addressed

2. Does this manuscript meet PLOS Global Public Health’s publication criteria? Is the manuscript technically sound, and do the data support the conclusions? The manuscript must describe methodologically and ethically rigorous research with conclusions that are appropriately drawn based on the data presented.

Reviewer #2: Partly

Reviewer #3: Yes

Reviewer #4: Yes

3. Has the statistical analysis been performed appropriately and rigorously?

Reviewer #2: Yes

Reviewer #3: Yes

Reviewer #4: Yes

4. Have the authors made all data underlying the findings in their manuscript fully available (please refer to the Data Availability Statement at the start of the manuscript PDF file)?

Reviewer #2: Yes

Reviewer #3: Yes

Reviewer #4: Yes

5. Is the manuscript presented in an intelligible fashion and written in standard English?

Reviewer #2: Yes

Reviewer #3: Yes

Reviewer #4: Yes

6. Review Comments to the Author

Reviewer #2: Thank you for the great improvement and work done thus far.

However, kindly pay attention and modify these few observations.

"Line 107" - Provide a reference to the statement made about Wakiso district.

"Line 132" - Reference this ((www.random.org?)) appropriately according to the publication criteria.

"Line 137" - Clarify from which study participants (humans or animals) were you checking for the history of diarrhea episodes and specify how you ascertained the information.

"Line 225, 271, 280, 303" - Clarify the error response feedback (Error! Reference source not found.).throughout the paper.

"Line 272, 304" - Clarify the table error response (Table Error! Reference source not found), found throughout the manuscript.

Reviewer #3: 1, Line 298 and 301still lokks like repetition. It might be scientifically correct but same aOR looks odd.

2. Except for table 1,All table titles (and the line before it) have not be correctly written. –Contains the word *Error* - probably the author doesn’t see this and we only seeit during conversion to pdf

Reviewer #4: Please edit every aspect of the manuscript where the words "source file not found" are appearing.

Though the authors have addressed the majority of the previous comments raised, I still feel strongly that there is a need for the limitation associated with this study be presented just before the conclusion. For instance, the authors did not characterize the ESBL isolates using molecular tools that would allow for adequate discrimination of the strains. This limitation, among others should be discussed.

7. PLOS authors have the option to publish the peer review history of their article (what does this mean?). If published, this will include your full peer review and any attached files.

**Do you want your identity to be public for this peer review?** For information about this choice, including consent withdrawal, please see our Privacy Policy.

Reviewer #2: No

Reviewer #3: No

Reviewer #4: No

---

## [Editor Report · Decision Letter 3]

2 May 2023

Epidemiology of Extended-spectrum beta-lactamase-producing Escherichia coli at the human-animal-environment interface in a farming community of central Uganda

PGPH-D-22-01776R3

Dear Mr Muleme,

We are pleased to inform you that your manuscript 'Epidemiology of Extended-spectrum beta-lactamase-producing Escherichia coli at the human-animal-environment interface in a farming community of central Uganda' has been provisionally accepted for publication in PLOS Global Public Health.

Best regards,

Ismail Ayoade Odetokun, DVM, Ph.D.

Academic Editor

Please attend to the editing errors noticed by the reviewers.